# Recent Advancements in the Utilization of *s*-Block Organometallic Reagents in Organic Synthesis with Sustainable Solvents

**DOI:** 10.3390/molecules29071422

**Published:** 2024-03-22

**Authors:** María Jesús Rodríguez-Álvarez, Nicolás Ríos-Lombardía, Sergio E. García-Garrido, Carmen Concellón, Vicente del Amo, Vito Capriati, Joaquín García-Álvarez

**Affiliations:** 1Dipartimento di Farmacia—Scienze del Farmaco, Università degli Studi di Bari Aldo Moro, Consorzio Interuniversitario Nazionale “Metodologie e Processi Innovativi di Sintesi” (C.I.N.M.P.I.S.), Via E. Orabona 4, I-70125 Bari, Italy; 2Laboratorio de Química Sintética Sostenible (QuimSinSos), Departamento de Química Orgánica e Inorgánica, Instituto Universitario de Química Organometálica “Enrique Moles” (IUQOEM), Facultad de Química, Universidad de Oviedo, E-33071 Oviedo, Spain

**Keywords:** organolithium, organosodium, organomagnesium, green chemistry, deep eutectic solvents, water

## Abstract

This mini-review offers a comprehensive overview of the advancements made over the last three years in utilizing highly polar *s*-block organometallic reagents (specifically, RLi, RNa and RMgX compounds) in organic synthesis run under bench-type reaction conditions. These conditions involve exposure to air/moisture and are carried out at room temperature, with the use of sustainable solvents as reaction media. In the examples provided, the adoption of *Deep Eutectic Solvents* (*DESs*) or even water as non-conventional and protic reaction media has not only replicated the traditional chemistry of these organometallic reagents in conventional and toxic volatile organic compounds under Schlenk-type reaction conditions (typically involving low temperatures of −78 °C to 0 °C and a protective atmosphere of N_2_ or Ar), but has also resulted in higher conversions and selectivities within remarkably short reaction times (measured in s/min). Furthermore, the application of the aforementioned polar organometallics under bench-type reaction conditions (at room temperature/under air) has been extended to other environmentally responsible reaction media, such as more sustainable ethereal solvents (e.g., CPME or 2-MeTHF). Notably, this innovative approach contributes to enhancing the overall sustainability of *s*-block-metal-mediated organic processes, thereby aligning with several key principles of *Green Chemistry*.

## 1. Introduction

Highly polar organometallic reagents based on *s*-block elements, basically organolithium (RLi) [1,2,3,4,5], organosodium (RNa) [6,7,8,9] and organomagnesium (RMgX or R_2_Mg) [10,11,12] compounds, stand out as some of the most reliable and extensively employed reagents in organic synthesis. Among these polar organometallic compounds, RLi reagents have played a pivotal role as indispensable tools since their discovery by Schlenk and Holz in 1917 [13]. Following this groundbreaking work, and owing to the contributions of other pioneers in the field (such as Ziegler [14], Wittig [15] and Gillman [16,17]), continuous advancements in the synthesis and use of these highly reactive reagents have expanded their applications across various domains of synthetic organic chemistry. These include lithium–halogen exchange processes, metalation reactions, Pd-catalyzed C–C coupling protocols and anionic polymerization reactions [1,2,3,4,5]. Consequently, it is not surprising that over 95% of methodologies employed in the total syntheses of natural products rely on RLi reagents in at least one step [5].

The attractiveness of RLi reagents to synthetic organic chemists lies in their (*i*) high and rich reactivity, (*ii*) commercial availability (typically sold as hydrocarbons or ethereal solutions) and (*iii*) affordable prices. However, their inherent high reactivity also presents certain challenges, primarily related to the tendency of RLi compounds to react with organic solvents, thereby requiring the use of low temperatures (ranging from 0 to −78 °C). Notably, it is well documented that THF undergoes decomposition (reverse cycloaddition reaction) in the presence of *n*-BuLi at room temperature [18]. For even more reactive *s*-BuLi or *t*-BuLi, this decomposition reaction occurs much more rapidly, thereby requiring storage/manipulation at temperatures below 0 °C [19]. Furthermore, owing to the high polarity of the C–M bond (M = Li, Na or Mg), additional side reactions, such as hydrolysis/oxidation, become prominent. Consequently, the use of these polar organometallic compounds requires (*i*) the use of rigorously dried, aprotic, toxic and volatile organic compounds (*VOCs*) as solvents and (*ii*) an inert atmosphere (usually N_2_ or Ar) to prevent side reactions in the presence of moisture or oxygen.

Contrary to conventional wisdom, recent innovations in *s*-block polar organometallic chemistry [20,21,22,23,24,25,26,27] have challenged established norms. It has been demonstrated that polar organolithium (RLi) and Grignard (RMgX) reagents can be effectively used (*i*) in the presence of air; (*ii*) at room temperature, and (*iii*) in sustainable, protic, non-toxic and non-dried solvents, such as water or *Deep Eutectic Solvents* (*DESs*), which are eutectic mixtures of Lewis or Brønsted acids and bases strongly associated with each other, exhibiting a significant depression of freezing points far below that of ideal mixtures (each component has a higher melting point than the mixture) [28]. This groundbreaking approach not only reduces costs, but also minimizes waste generation and energy consumption, contributing to a more sustainable use of fossil resources. Furthermore, the application of aerobic and non-dried reaction conditions for RLi/RMgX has also been reported in flow processing [29,30,31,32,33] and solid-phase synthesis [34]. Additionally, the use of more sustainable ethereal solvents (e.g., 2-methyltetrahydrofuran (2-MeTHF) or cyclopentyl methyl ether (CPME) [35]) has broadened the possibilities for employing these common s-block metal reagents under atmospheric conditions and in the presence of moisture. In this sense, 2-MeTHF is considered a biorenewable biomass-derived solvent, as it can be produced from furfural without the need for petrochemical protocols [36]. Moreover, the immiscibility between water and 2-MeTHF permits its straightforward employment in liquid–liquid extraction (avoiding the use of toxic *VOC* solvents for the purification/isolation of the desired final organic products), thus opening the door to its use as an alternative solvent in organometallic-mediated organic transformations [37]. On the other hand, CPME is also considered a promising ecofriendly solvent due to its intrinsic valuable properties such as a low peroxide formation rate, hydrophobicity, stability under basic and acidic conditions and relatively high boiling point [35].

At this point, it is crucial to highlight that, beyond the sustainable perspective, the utilization of bench-type reaction conditions (presence of air/moisture at room temperature) in synergistic conjunction with the aforementioned unconventional, protic and polar reaction media (water and *DES*) or sustainable ethereal solvents (2-MeTHF/CPME) in polar organometallic chemistry has yielded significant advantages. This approach has, indeed, enabled (*i*) notable improvements in reaction times (on the scale of s/min); (*ii*) enhanced yields/selectivities in well-established chemical transformations and (*iii*) the discovery of novel reactions/mechanistic pathways that cannot be replicated in traditional, dry and toxic volatile organic compounds (*VOCs*) as solvents [20,21,22,23,24,25,26,27].

In the case of protic polar solvents (water and *DESs*), these empirical observations have so far been connected to the exceptional chemical and physical properties of these non-conventional solvents, which can exert a strong influence on (*i*) the position of chemical equilibria, (*ii*) the reaction rates in both heterogeneous and homogeneous protocols, and (*iii*) the overall outcome of the synthetic organic process under study [38,39,40,41,42,43,44,45,46,47,48,49,50,51,52]. In this context, a thorough review of the scientific literature, focusing specifically on the pair formed by RLi/RMgX reagents and water, unequivocally demonstrates that the intentional (or adventitious) addition of catalytic or stoichiometric amounts of water favors various main group-mediated organic protocols by (*i*) speeding up the desired reaction [53], (*ii*) enhancing a lithium/halogen exchange protocol [54,55] or (*iii*) improving the enantiomeric excess in asymmetric protocols [56].

With this idea in mind, our focus in this review is on recent advancements (since 2021) in the use of *s*-block polar organometallic reagents under aerobic/ambient conditions when employing sustainable reaction media such as *DESs*, water, 2-MeTHF or CPME as solvents. This review is structured into distinct sections, each highlighting various types of reactivities: (*i*) chemoselective double addition of RLi/RMgX reagents into esters in *DESs* or water at room temperature and in the absence of a protective atmosphere [57]; (*ii*) the potential implementation of RLi/RMgX reagents under continuous, stable and safe reaction conditions in flow at room temperature, assisted by *DESs* or water [58,59]; (*iii*) the rapid addition of *s*-block organometallic reagents to CO_2_-derived cyclic carbonates at room temperature, under air and in 2-MeTHF [60]; (*iv*) the one-pot/two-step modular double addition of different highly polar organometallic reagents (RLi/RMgX) into nitriles to produce asymmetric tertiary alcohols under aerobic/room temperature and in neat conditions [61]; (*v*) the fast addition of RLi reagents to amides in environmentally friendly CPME, at ambient temperature and under air [62]; (*vi*) the highly efficient and selective fast addition of in situ generated lithium amides (obtained via an acid–base reaction between *n*-BuLi and the desired primary amine) into carbodiimides (R-N=C=N-R) or nitriles (R-C≡N) in 2-MeTHF or CPME [63]; (*vii*) the Directed *ortho*-Metalation (D*o*M) or anionic Fries rearrangement of *O*-arylcarbamates promoted by lithium amides under aerobic conditions in sustainable reaction media (*DESs*/CPME) [64]; (*viii*) the Pd-catalyzed cross-coupling reactions of RLi reagents with *o*-benzenedisulfonimides [65]; (*ix*) the electrophilic trapping of in situ generated organosodium intermediates (RNa) in *DESs* or in pure water, at room temperature and under air [66], and (*x*) the neopentyl sodium (NpNa)-promoted sodium–halogen exchange of aromatic products or direct deprotonation of terminal alkynes under mild conditions [66]. Additionally, the final section of this review summarizes ongoing efforts to design hybrid one-pot tandem protocols aimed at synergistically combining aerobic polar s-block organometallic chemistry in green solvents with (*i*) enzymatic aqueous synthetic protocols, such as oxidations of secondary alcohols into ketones [67] or the biodeoximation of ketoximes [68]; (*ii*) organic procedures, including the acetylation of diols [69] or tetrahydropyranylation of alcohols, promoted by Brønsted-type acidic *DES*s [70] and (*iii*) the Cu-catalyzed oxidation of primary alcohols into the corresponding aldehydes in *DESs* or water [71].

## 2. Addition of RLi/RMgX Reagents into Different Electrophiles in Green Solvents

A groundbreaking study conducted by the Hevia and García-Álvarez research teams in 2014 [72] has brought forth a novel perspective on the potential utilization of highly polar *s*-block organometallic reagents (RLi/RMgX) within more environmentally friendly reaction conditions. Specifically, the feasibility of conducting reactions at room temperature in the presence of air and moisture, achieved through the use of choline chloride (*ChCl*)-based eutectic mixtures as green and innovative solvents, was explored [73]. Our investigation demonstrated that these *DESs* promoted ultrafast (2–3 s) and chemoselective additions of organometallic reagents into ketones. By comparing the reactivity of RMgX and RLi reagents in these *ChCl*-based *DESs* with that observed in pure water as a reaction medium, a mechanism involving a kinetic activation was proposed. It most probably involves the formation of highly nucleophilic and halide-rich anionic magnesiate ([RMgCl_2_]^−^) or lithiate ([LiCl_2_R]^2−^) species, facilitating their nucleophilic addition into ketones while minimizing undesired side reactions such as enolizations and hydrolysis. Building upon this seminal example, the scope of unsaturated electrophiles, amenable to a fast and chemoselective addition in *DESs*, was then extended to include (*i*) imines [74,75], (*ii*) nitriles [76,77] and (*iii*) aldehydes/epoxides [78]. This expansion broadens the potential applications of *DESs* as green solvents in organic synthesis, opening avenues for sustainable and efficient reactions across diverse substrates.

### 2.1. Chemoselective Addition of Polar Organometallic Reagents (RLi/RMgX) to Esters under Air at Room Temperature and Using Either Water or ChCl-Based Eutectic Mixtures as Reaction Media

Recently, our research endeavors have expanded upon the previously discussed nucleophilic addition of RLi/RMgX reagents in green solvents, leading to the development of a novel, straightforward and highly efficient synthetic protocol. This protocol enables the production of symmetric tertiary alcohols (**2**) with exceptional yields reaching up to 95%. The method involves a rapid (20 s) and chemoselective double addition of various highly polar *s*-block reagents (RLi/RMgX) to esters **1**, working at room temperature and under ambient air conditions. Notably, the biorenewable 1*ChCl*/2Urea eutectic mixture or pure water are employed as sustainable reaction media [57] (Figure 1).

It is crucial to highlight that this methodology exhibits several key features: (*i*) it demonstrates a broad scope, accommodating both aromatic and aliphatic substituents in the starting ester **1**, as well as a variety of organometallic reagents (including *n*-BuLi, *i*-PrLi, PhLi, 2-ThienylLi, MeLi, PhMgCl, MeMgCl or EtMgBr); (*ii*) it obviates the need for isolating intermediate ketones, streamlining the synthetic pathway; and (*iii*) it proves applicable to the synthesis of *A*ctive *P*harmaceutical *I*ngredient (*API*)-type organic products, exemplified by the synthesis of *S*-trytil-L-cysteine—an influential Eg5 inhibitor. This advancement underscores the versatility and practicality of our synthetic approach, contributing to the sustainable and efficient synthesis of valuable organic compounds [57].

### 2.2. Continuous, Stable and Safe Organometallic Reactions in Flow at Room Temperature Assisted by Deep Eutectic Solvents

Continuous flow technologies have gained widespread acceptance in the realm of organic chemistry, emerging as a highly dependable synthetic approach, particularly when dealing with highly reactive reagents in both laboratory and industrial settings [30,31,32,33,79,80]. Elevating this concept to new heights, Hevia, Torrente-Murciano et al. presented a groundbreaking contribution [58] by introducing the first continuous, stable and secure synthetic operation employing RLi/RMgX reagents in flow under ambient conditions with remarkable moisture tolerance and resistance to clogging. A contextualization and summary of the key aspects of this paper has also been published [81].

This innovation capitalizes on the utilization of two distinct *ChCl*-based eutectic mixtures, namely 1*ChCl*/2Urea and 1*ChCl*/2*Gly* (*Gly* denotes glycerol). Notably, the implementation of these eutectic mixtures eliminates the need for temperature control or the use of an inert atmosphere. Through the integration of a multi-phase, segmented-flow microfluidic system (depicted in Figure 1), the researchers observed a significant enhancement in mixing and heat transfer. This improvement is attributed to secondary vortex structures within the droplets and the high surface area-to-volume ratio, resulting in stable and secure operations. This novel protocol was successfully applied to the synthesis of alcohols and amines through the addition of various organometallic species to different ketones and imines, achieving yield values up to 98%. The adaptability and efficiency demonstrated by this continuous flow method mark a significant stride in the advancement of synthetic methodologies, offering a robust and reliable approach for diverse chemical transformations. In 2024, Benaglia and co-workers advanced this field by developing a highly efficient and rapid (10–20 s) continuous on-water organolithium addition to imines. This innovative approach allows for the synthesis of functionalized secondary amines in high yields (up to 97%), effectively addressing the safety concerns associated with the process. To achieve these impressive results, the authors employed continuous-stirred tank reactors (CSTRs) ensuring a rapid stirring rate. Notably, within a single 2.5 mL CSTR, 5 g of amine can be synthesized in just 3 min. Furthermore, the methodology has successfully been exploited for the synthesis of an enantiomerically enriched chiral amine (98% enantiomeric excess) [59].

### 2.3. Fast Addition of s-Block Organometallic Reagents to CO_2_-Derived Cyclic Carbonates at Room Temperature, under Air and in 2-MeTHF

In a collaborative effort involving some of us and Elorriaga, Castro-Osma et al. [60], we drew inspiration from an earlier refined method reported by Jessop, Snieckus and their colleagues. This method, focused on the conversion of C1 feedstock (specifically, sodium methyl carbonate), utilizing *s*-block polar organometallic reagents (RLi/RMgX) in dry organic solvents under Schlenk-type conditions with extended reaction times (24 h) [82]. Building on this foundation, we decided to explore the use of cyclic carbonates (derived from CO_2_, a sustainable feedstock) as biorenewable electrophiles capable of undergoing the ultrafast (3 s) addition of RMgX/RLi reagents, as illustrated in Figure 2.

Remarkably, our investigations revealed that, under optimal conditions—employing 2-MeTHF as a sustainable reaction medium or working in the absence of external *VOCs* (neat conditions)—and conducting reactions at room temperature in the presence of air and moisture, cyclic carbonates **3** could be efficiently converted into tertiary alcohols **4**, symmetric ketones **5** or *β*-hydroxy esters **6** through the direct addition of polar organometallic alkylating or arylating reagents (RMgX/RLi). This innovative approach not only significantly shortens reaction times, but also demonstrates the versatility of cyclic carbonates as renewable electrophiles in the synthesis of diverse chemical products.

In the same study, we successfully devised a hybrid one-pot/two-steps tandem protocol [83]. This innovative approach seamlessly integrates the aluminum-catalyzed insertion of CO_2_ into propylene epoxide (**7**) with the subsequent chemoselective and rapid addition of an organolithium reagent (EtLi) to the transiently formed cyclic carbonate **3a**. Notably, this protocol eliminates the need to isolate or purify the reaction intermediate [84], thus representing a direct and sustainable method for converting CO_2_ into tertiary alcohol **4a** (Figure 3).

### 2.4. One-Pot/Two-Step Modular Double Addition of Different Highly Polar Organometallic Reagents (RLi/RMgX) into Nitriles to Produce Asymmetric Tertiary Alcohols under Aerobic/Room Temperature and in Neat Conditions

In the pursuit of designing one-pot/two-step protocols leveraging *s*-block organometallic reagents (RLi/RMgX) under bench-type reaction conditions (room temperature and without the need for a protective atmosphere) with an emphasis on neat conditions (*the best solvent is no solvent*) [85], our collaborative efforts, led by Elorriaga, Carrillo-Hermosilla, and some of us, drew inspiration from the pioneering work of Capriati and colleagues that had previously demonstrated the double addition of RLi/RMgX reagents into nitriles, yielding tertiary carbinamines in water at room temperature [75]. Building on our prior knowledge of the RLi-mediated conversion of nitriles into ketones under air/moisture and at room temperature [77], we decided to revisit and expand this field.

Our newly developed one-pot modular tandem protocol began with the fast and chemoselective addition of various RLi reagents (ranging from aliphatic and silylated to aromatic or heteroaromatic groups) to a diverse array of nitriles (**8**, accommodating both aromatic and aliphatic substituents). This process took place under bench-type and neat conditions, eliminating the need for external *VOCs*. This first step generated the corresponding lithiated imines **9**, which, following acidic hydrolysis (using a saturated NH_4_Cl aqueous solution), yielded the desired ketones **10**.

Significantly, without any intermediate isolation or purification, the second step of our protocol facilitated the addition of a second RLi reagent to ketones **10**. This occurred even in the presence of an acidic reaction medium, at room temperature and under ambient air (Figure 4). Employing this innovative tandem methodology, we achieved the fast, direct and selective conversion of nitriles into the corresponding tertiary alcohols **11** in moderate to excellent yields [61]. Notably, we successfully demonstrated the scalability of this protocol, establishing it as an environmentally friendly synthetic tool for the efficient synthesis of non-symmetric tertiary alcohols from nitriles. This process operates under bench-type reaction conditions and avoids the use of external organic solvents.

### 2.5. Addition of Organolithium Reagents (RLi) into Amides as a General and Fast Route to Ketones Using CPME as a Sustainable Solvent under Aerobic Ambient Conditions

In their pioneering investigation focusing on the application of *t*-BuLi to facilitate the D*o*M of hindered benzamides under bench-type reaction conditions (room temperature and in the presence of air), and utilizing 1*ChCl*/2*Gly*:CPME mixtures as sustainable reaction media [86], Capriati, Prandi, Blangetti and co-workers found that substituting *t*-BuLi with less-hindered organometallic reagents, which are more inclined to induce addition reactions rather than the metalation process, led to the synthesis of aromatic ketones alongside the corresponding di-*n*-butylated symmetric tertiary alcohol—a by-product anticipated in a double addition protocol [87].

Building upon this experimental insight, the same researchers embarked on a systematic investigation to devise a novel synthetic protocol enabling the chemoselective conversion of amides **12** into the desired aromatic ketones **13** using polar organometallic reagents. This process was designed to operate under bench conditions, employing CPME as a sustainable solvent [62]. The authors discovered that the nucleophilic acyl substitution reaction (S_N_Ac) of aliphatic and (hetero)aromatic amides **12** could be efficiently promoted by various organolithium reagents, such as *n*-BuLi, *s*-BuLi, *n*-HexylLi, PhLi and 2-ThienylLi (Figure 5). This nucleophilic substitution takes place rapidly (within 20 s), with good product yields (up to 93%), and exhibits high chemoselectivity, avoiding the formation of over-addition by-products. Notably, this methodology demonstrates a remarkable tolerance for a broad substrate scope of substituents. It is also worth mentioning that (*i*) both DFT calculations and NMR investigations substantiated the observed experimental results, (*ii*) CPME and the leaving group were demonstrated to be reusable, and (*iii*) the reported methodology proved to be scalable, thereby enhancing its practical applicability.

### 2.6. Fast and Selective Addition of In Situ Generated Lithium Amides (LiNR_2_) into Carbodiimides (R-N=C=N-R) or Nitriles (R-C≡N) in 2-MeTHF or CPME under Air and at Room Temperature

In the preceding sections of this review, the examples discussed primarily involved the formation of new C–C bonds through the fast and selective addition of RLi/RMgX reagents to various organic electrophiles, working at room temperature and under ambient air conditions [57,58,60,61,62]. However, in previous collaborative efforts with Hevia and co-workers, some of us reported on the synthesis of new C–N bonds using lithium amides (LiNR_2_) as nucleophiles. Specifically, we explored (*i*) the direct transamidations of esters or highly activated *N*-Boc-protected amides [88], and (*ii*) the moisture-promoted hydroamination of vinylarenes [89]. Notably, both reactions were performed under aerobic ambient temperature conditions, employing 2-MeTHF as a sustainable solvent.

Building upon this chemistry, Elorriaga, Carrillo-Hermosilla, and some of us, then extended these investigations to the addition of in situ generated lithium amides LiNHR (prepared via an acid–base reaction between *n*-BuLi and the desired primary amines RNH_2_ **14**) into two distinct nitrogenated and unsaturated organic electrophiles—carbodiimides (**15**; R-N=C=N-R) or nitriles (**16**; R-C≡N). This extension was carried out at room temperature, without the need for a protective atmosphere, and utilizing either 2-MeTHF or CPME as sustainable solvents [63].

For carbodiimides **15**, optimal yields of the desired guanidines **17** were achieved when 2-MeTHF was employed as the solvent. Conversely, the use of CPME was essential to obtain the corresponding amidines **18** when using aromatic nitriles **16** as electrophiles (Figure 6). Importantly, our one-pot/two-step synthetic protocols demonstrated versatility in accommodating various functional groups, present in both the starting amine **14** and the electrophiles **15**,**16**. This approach resulted in a diverse array of *N*-containing and highly substituted organic products **17**,**18** obtained in good to excellent yields and under more environmentally friendly reaction conditions.

## 3. Directed *ortho*-Metalation or Anionic Fries Rearrangement of *O*-Arylcarbamates Promoted by Lithium Amides under Aerobic Conditions in Sustainable Reaction Media (*DESs*/CPME)

Despite the notable advancements in the rapid and selective nucleophilic addition of RLi/RMgX reagents into various unsaturated organic electrophiles using sustainable solvents and bench-type reaction conditions [57,58,60,61,62,63,64], the exploration of the behavior of RLi/RMgX reagents as bases in greener reaction media is still in its early stages. The limited research in this area can be attributed to the conventional wisdom that polar organolithium or organomagnesium compounds would readily undergo protonation in the presence of unconventional protic solvents like water or *DESs*.

Although the nucleophilic character of RLi or RMgX reagents in *ChCl*-based *DESs* can be augmented through two strategies: (*i*) co-complexation with an ammonium salt, leading to the formation of the corresponding anionic lithiates ([LiCl_2_R]^2−^) or magnesiates ([RMgX_2_]^−^), and/or (*ii*) hydrogen bond interactions within a heterogeneous mixture under vigorous stirring, the classical deprotonation/electrophilic trapping protocol involves the generation of a highly reactive carbanionic intermediate (C-Li), which requires a proton-free environment to prevent undesired re-protonation reactions, leading to the recovery of the unreacted starting material.

The initial hurdle of working with protic *DESs* was successfully overcome by Capriati and co-workers by (*i*) achieving the D*o*M of diaryltetrahydrofurans, followed by trapping with various electrophiles [90], and (*ii*) running a one-pot tandem lateral lithiation/alkylative ring opening of *o*-tolyltetrahydrofurans [91], both conducted in the presence of protic *DESs* as the reaction medium. Building on these pioneering efforts, Prandi, Blangetti and co-workers expanded the scope of lithiation/electrophilic trapping in the presence of *DESs* to (*i*) the aforementioned D*o*M and subsequent electrophilic trapping of hindered carboxamides [86], and (*ii*) the heteroatom-directed lateral lithiation of substituted toluene derivatives [92].

Subsequently, the same research team delved into *N*,*N*-dialkyl-*O*-phenylcarbamates, revealing an intriguing dichotomy in the behavior of these aromatic organic products based on (*i*) the reaction temperature (0 °C or room temperature), (*ii*) the nature of the organolithium reagents used (LiTMP vs. *s*-BuLi), and (*iii*) the reaction media employed (pure CPME vs. mixtures of CPME/*DESs*) [64]. Aromatic carbamates are conventionally recognized as excellent directors for D*o*M protocols, enabling clean metalation at their *ortho*-position at low temperatures. This facilitates their chemoselective and straightforward functionalization after the corresponding electrophilic quenching [93]. Additionally, in the absence of an external electrophile, the aryl anion undergoes a rapid intramolecular carbamoyl transfer upon gradual warming to room temperature, a process known as anionic *ortho*-Fries rearrangement [94]. This process yields the corresponding salicylamides, which can also be utilized in subsequent D*o*M protocols. In light of these considerations, Prandi and co-workers found that the optimal conditions for a selective D*o*M functionalization of phenylcarbamate **19** (avoiding Fries rearrangement) involves (*i*) working at low temperatures (0 °C) in the presence of air/moisture, (*ii*) using *s*-BuLi as the organolithium reagent, and (*iii*) employing an equimolecular mixture of the eutectic mixture 1*ChCl*/2*Gly* and CPME (Figure 7). This results in the formation of the corresponding *ortho*-functionalized carbamates **20** in moderate to good yield. Lastly, it is noteworthy that this protocol allows for the use of different trapping electrophiles, leading to the formation of various C–C or C–heteroatom bonds [86].

In their comprehensive study [64], Prandi et al. successfully identified optimized reaction conditions for achieving a chemoselective anionic *ortho*-Fries rearrangement in a series of phenyl *N*,*N*-diisopropyl and *N*,*N*-diethyl carbamates **21**, encompassing a diverse range of functionalities on the aromatic ring (Figure 8). The Fries-type reaction conditions consistently involved (*i*) the use of CPME as a solvent, thereby excluding *DESs*, (*ii*) operation under ambient air conditions at room temperature and (*iii*) utilizing a freshly prepared solution of LiTMP (TMP = 2,2,6,6-tetramethylpiperidine) as the metalating agent, rather than carbanionic RLi reagents. Employing this set of conditions yielded the desired salicylamides **22** in nearly quantitative yields within remarkably short reaction times (60 s). Particularly noteworthy is the practical applicability of the authors’ protocol in the pharmaceutical industry, as demonstrated by the synthesis of (*i*) an acetylcholinesterase inhibitor (*AChEI*) for Alzheimer’s disease treatment, and (*ii*) the blockbuster drug acetaminophen (Paracetamol).

## 4. Pd-Catalyzed C–C Coupling Reactions Using RLi Reagents in Green Solvents

Since Murahashi’s groundbreaking work on the potential application of direct organolithium-promoted cross-coupling reactions in organic synthesis [95], RLi reagents have found extensive utility as coupling partners in Pd-catalyzed C–C processes. However, the adoption of these RLi organometallic compounds in C–C cross-coupling protocols in environmentally friendly and protic solvents has long been overlooked by synthetic organic chemists. This neglect stemmed from the perceived high susceptibility of these polar reagents to hydrolysis reactions in protic media.

In 2016, Feringa and co-workers reported a chemoselective Pd-catalyzed C–C coupling protocol involving PhLi (2–10 equiv) and 1-bromonaphthalene as partners, utilizing a protic *DES* as the reaction medium at room temperature. Under these conditions, the final C–C coupling product was obtained in moderate to good yields (28–53%) and within a short reaction time (10 min) [96]. Building upon this work, Capriati and co-workers then demonstrated the feasibility of promoting Pd-catalyzed C(sp^3^)–C(sp^2^), C(sp^2^)–C(sp^2^) and C(sp)–C(sp^2^) cross-coupling reactions between RLi reagents and a variety of (hetero)aryl halides (Cl, Br) that occurred competitively with protonolysis when using water as a non-innocent reaction medium (*on-water* conditions) and NaCl as a cheap additive. All the reactions proceeded within a reaction time of 20 s, in yields of up to 99%, at room temperature and under ambient air conditions. The proposed protocol was also scalable, and the catalyst and water could easily be reused up to 10 times, with an *E*-factor as low as 7.35 [97].

In a more recent development in 2023, Dughera and Antenucci expanded the application of RLi reagents in protic solvents for the study of Pd-catalyzed coupling reactions involving the use of arenediazonium *o*-benzenedisulfonimides **23** (instead of the traditionally used aryl halides) and aryl/heteroaryl organolithium reagents. Reactions were conducted in the eutectic mixture formed by KF and glycerol (1KF/6*Gly*) using Pd[P(*t*-Bu)_3_]_2_ as the catalyst (Figure 9). Operating at room temperature and in the presence of air, the desired biaryl products **24** were obtained after short reaction times (5 min) and with moderate to good yields (27–80%) [65].

## 5. Organosodium Chemistry (RNa) in Green Solvents

Organosodium reagents (RNa) stand out for their highly reactive Na–C bonds, potentially rivaling or even surpassing the reactivity of the previously mentioned organolithium reagents [6,7,8,9]. However, their increased reactivity often encounters significant challenges, including poor solubility in common organic solvents and limited stability. These limitations have historically restricted their application in organic synthesis. Nevertheless, in recent years, the resurgence of RNa-promoted organic synthesis can be attributed to the integration of sustainability concepts, such as the fact that sodium is more abundant (earth crustal abundance 22,700 ppm) than lithium (abundance 18 ppm) [9], thus reigniting the interest in sodium-promoted synthetic chemistry.

Capitalizing on the heightened reactivity and distinctive regioselectivities observed for RLi/RMgX reagents in protic and non-conventional solvents, such as water or eutectic mixtures, under ambient conditions and in the presence of air, Capriati and co-workers sought to extend this chemistry to organosodium reagents (RNa) [66]. To achieve this goal, the authors devised a method to generate highly reactive C(sp^3^), C(sp^2^) and C(sp) organosodium reagents in situ using hexane as a solvent. This synthetic protocol relies on the oxidative addition reactions of C–Cl bonds to sodium, originating from both aryl and alkyl organochlorides (R-Cl, **25**), and utilizes sodium bricks (*SBs*), small pieces of sodium cut from larger lumps. The resulting organosodium intermediates, denoted as RNa (**26**), were then promptly (within 20 s) subjected to an electrophilic trapping protocol.

This trapping reaction successfully took place in two distinct *DESs*—namely, 1*ChCl*/2Urea and 1*Pro*/3*Gly* (*Pro* = L-proline)—or even in pure water. All reactions were run at room temperature and in the presence of air/moisture, yielding the final organic products **27** in good to almost quantitative yields (58–98%). Importantly, the electrophilic quenching process was found to be favored over the undesired competing hydrolysis reaction, as shown in Figure 10 [66]. This innovative approach opens new avenues for the application of organosodium reagents in more sustainable organic syntheses, bridging the gap between their high reactivity and the challenges associated with their practical utility.

Furthermore, within the same study [66], Capriati and co-workers employed the previously optimized procedure to synthesize neopentyl sodium (NpNa; **29**) through the oxidative addition reaction involving the C–Cl bond of neopentyl chloride (NpCl; **28**) and the aforementioned *SBs* in hexane. The in situ generated **29** exhibited remarkable versatility, facilitating either sodium–halogen exchange reactions with aryl bromides (**30**) or the direct deprotonation of terminal alkynes (**31**) under mild conditions (at 0 °C in hexane; Figure 11). The resulting organosodium species **32**,**33** could then be easily intercepted using a variety of simple electrophiles, either in water or in the eutectic mixture 1*ChCl*/2Urea, with all reactions performed at room temperature and in the presence of air.

## 6. One-Pot Tandem Hybrid Combinations of RLi Reagents with Other Synthetic Organic Tools in Water or *Deep Eutectic Solvents* (*DESs*)

The findings presented in this review challenge the conventional notion that organolithium (RLi), organosodium (RNa) or Grignard (RMgX) reagents must be exclusively handled under stringent Schlenk-type conditions and in the absence of protic solvents [57,58,60,61,62,63,64,65,66]. This paradigm shift creates new opportunities for developing innovative and environmentally friendly one-pot tandem hybrid protocols under greener reaction conditions utilizing sustainable and protic reaction media. These tandem hybrid protocols hold the promise of seamlessly integrating highly efficient main group-mediated organic transformations, run in protic and unconventional solvents, with other synthetic techniques available to organic chemists, such as homogeneous catalysis promoted by transition metals, enzymes or organocatalysts. For instance, some of us have previously demonstrated the feasibility of combining the addition of RLi/R_2_Mg reagents with (*i*) the Ru(IV)-catalyzed redox isomerization of allylic alcohols in *DESs* [98], and (*ii*) the organocatalytic oxidation of secondary alcohols into ketones, a process performed in aqueous media [99].

### 6.1. Design of One-Pot Hybrid Chemoenzymatic Protocols That Rely on the Use of RLi/RMgX Reagents in Aqueous Media

Following our previous success in combining RLi reagents with organocatalyzed oxidations of secondary alcohols in water under bench-type reaction conditions [99], we extended our investigation by introducing a novel chemoenzymatic hybrid one-pot tandem protocol. In this approach, the biocatalytic oxidation of secondary alcohols **36**, catalyzed by a commercially available laccase from *Trametes Versicolor* [100], was fruitfully integrated through a sequential one-pot/two-step procedure, eliminating the need for intermediate isolation/purification steps. This protocol was specifically designed to efficiently combine the rapid and selective addition of RLi reagents into transiently formed ketones **37** [67]. Notably, the addition of RLi reagents into ketones **37** occurs under biocatalytic conditions, involving (*i*) water/CPME mixtures as the reaction medium at room temperature, (*ii*) TEMPO (2,2,6,6-tetramethylpiperidine 1-oxyl) as a co-factor, and (*iii*) aerial O_2_ as a co-oxidant (Figure 12). Interestingly, the less reactive Grignard reagents (RMgX) underwent complete protonolysis under these conditions. Finally, the non-symmetric tertiary alcohols **38** were obtained in good to almost quantitative yields (65–96%).

After successfully integrating RLi reagents and laccases in aqueous media under bench-type reaction conditions, and considering the feasibility of *Trametes Versicolor* laccases to catalyze deoximation reactions in such an environment [101], the authors decided to further expand their investigation. In a sequential hybrid protocol, they combined the biocatalytic deoximation of aromatic ketoximes **39** with the subsequent addition of RLi reagents into the transiently formed ketones **40**. This innovative one-pot/two-step procedure takes place in aqueous media at room temperature and in the presence of air, yielding the desired tertiary alcohols **41** in moderate to good yields (46–88%). Importantly, this entire process eliminate the need for isolation or purification steps [68]. Moreover, this hybrid approach demonstrated exceptional adaptability, marking its capacity to accommodate less reactive Grignard-type organometallic reagents (RMgX), a novel feat in this domain of chemistry (Figure 13).

### 6.2. Combination of Brønsted–Acidic–DES-Promoted Organic Protocols with the Addition of RLi Reagents

Brønsted-type acidic *DESs* represent a versatile class of eutectic mixtures widely employed as promoters/catalysts in various organic transformations encompassing a broad spectrum, ranging from Claisen–Schmidt procedures and Friedel–Craft acylations to esterification reactions, biodiesel production, and multicomponent condensation reactions [102,103]. Notably, these *DESs* are composed by utilizing naturally occurring carboxylic acids as hydrogen bond donors. In this realm of research, Prandi, Blangetti and their co-workers have recently showcased the integration of Brønsted–acidic–*DESs* in the chemistry of protecting groups for aldehydes/ketones or alcohols. They have further demonstrated the feasibility of combining this chemistry with subsequent RLi-mediated synthetic transformations, thus delineating two distinct telescoped approaches for one-pot tandem procedures involving organolithium reagents under more sustainable conditions. In the first case, Prandi and co-workers demonstrated the effectiveness of using a malonic acid (*Mal*)-containing eutectic mixture (1*ChCl*/1*Mal*) as both an environmentally friendly solvent and a promoter for the selective acetalization reaction of diols with carbonyl compounds working at room temperature, in air and with short reaction times [69]. In this work, the authors chose methyl *p*-formylbenzoate **42** as the ideal substrate for the design of a hybrid tandem protocol (as it contains both aldehyde and ester functional groups in its structure) to synthesize the target compound **45**. In this approach, the more electrophilic aldehyde function needed protection towards the addition of *n*-BuLi. Thus, **42** was, firstly, converted into acetal **43** in the presence of neopentyl glycol by using (1*ChCl*/1*Mal*) as the promoter/solvent. The in situ obtained acetal **43** was directly utilized (without the need of any halfway isolation/purification protocol) in the subsequent step, which involved the addition of *n*-BuLi under ambient air conditions (CPME was added as a co-solvent to facilitate vigorous stirring) to produce the tertiary alcohol **44**. Finally, the desired product **45** was obtained just by a simple and straightforward acidic work-up with 61% overall yield over the three sequential steps (Figure 14).

Secondly, in a closely related study [70], the same authors expanded the scope of this telescoped chemistry by demonstrating the feasibility of combining the acidic–*DES*-promoted tetrahydropyranylation of primary alcohols (a process occurring under air, within short reaction times and exhibiting exceptional chemoselectivity) with the subsequent addition of RLi reagents in the presence of the acidic eutectic mixture. In this context, Prandi and co-workers outlined a hybrid one-pot tandem protocol for converting benzyl alcohol **46** into the desired hydroxylated valerophenone **50** (Figure 15). The initial step of this hybrid tandem transformation involves the conversion of alcohol **46** into the corresponding pyranyl derivative **48** by treating **46** with 3,4-dihydro-2*H*-pyran (**47**) in the eutectic mixture 1*ChCl*/1*Mal* at 50 °C and under air. The subsequent addition of *n*-BuLi to the in situ generated intermediate **48** in a heterogeneous CPME/*DES* mixture triggers the expected S_N_Ac, resulting in ketone **49**. After a final deprotection step, the corresponding valerophenone **50** is obtained with an overall yield of 57% (Figure 15). Importantly, this hybrid protocol was extended to other benzylic alcohols containing ester or nitrile groups in their structures.

### 6.3. Combination of a Cu-Catalyzed Oxidation of Primary Alcohols into Aldehydes with the Addition of RLi Reagents in DESs

The conceptualization of hybrid one-pot tandem protocols, combining transition metal-catalyzed organic processes with main group-mediated synthetic transformations, has garnered significant interest in the scientific community, as both methodologies represent fundamental pillars in the toolkit of synthetic organic chemistry. However, instances of these hybrid transformations utilizing protic and environmentally friendly reaction media, such as water and *DESs*, are still relatively scarce. In this context, some of us have previously reported the successful integration of the Ru(IV)-catalyzed redox isomerization of allylic alcohols into their corresponding carbonyl compounds with a concurrent, rapid and chemoselective addition of RLi/R_2_Mg reagents in *ChCl*-based eutectic mixtures, yielding their corresponding tertiary alcohols in a sequential tandem protocol [98]. In a subsequent study, Prandi, Capriati and co-workers successfully combined the lithiation–iodation process of aromatic amides with a consecutive Pd-catalyzed Suzuki–Miyaura cross-coupling reaction using various borates/boronic acids as coupling partners in mixtures of 1*ChCl*/2*Gly*:CPME [86].

In 2022, following these seminal studies, some of us explored the combination of the chemoselective oxidation of benzyl alcohol (**51**) into benzaldehyde (**52**) using the Cu(II)-based oxidative catalytic system CuCl_2_/TMEDA/TEMPO (TMEDA = *N*,*N*,*N*’,*N*’-tetramethylethylenediamine) under sonication (US). This was followed by the subsequent addition of RLi/RMgCl reagents, run in the presence of air and employing water or the eutectic mixture 3*Fruc*/2Urea (*Fruc* = D-fructose) as more sustainable reaction media [71]. Through this hybrid protocol, the primary alcohol **51** could be directly transformed into secondary alcohols **53** in moderate to good yields, using bench-type reaction conditions and without the need for any intermediary isolation/purification steps (Figure 16).

## 7. Conclusions

Polar organometallic reagents belonging to the *s*-block elements (RLi, RNa or RMgX) play a crucial role in synthetic chemistry, being widely employed for the functionalization of organic molecules owing to the high polarity of their M–C bonds. However, their application is typically constrained by the requirement for low temperatures, the use of dry, toxic organic solvents, and the need for protective inert atmospheres to prevent rapid decomposition. Traditionally, the challenge of conducting organolithium, organosodium or organomagnesium chemistry, under aerobic and/or hydrous conditions and in the absence of dry toxic solvents, has been considered insurmountable. Nevertheless, this review showcases that recent works from various research groups worldwide are expanding the field of research related to the use of highly polar organometallic reagents under bench-type reaction conditions (room temperature and in the absence of a protective atmosphere) and in more sustainable reaction media, such as water, Deep Eutectic Solvents (*DESs*) and greener ethereal solvents (2-MeTHF or CPME). Furthermore, the possibility of integrating these methods with enzymes or transition metal catalysis introduces an array of opportunities for designing innovative hybrid one-pot protocols under environmentally friendly reaction conditions. Thus, the progress compiled in this review represents a substantial advance toward making polar organometallic chemistry more sustainable and provides a versatile platform for diverse and greener synthetic pathways.

The examples reviewed not only underscore the sustainability of the transformations, but also reveal that the exceptional chemoselectivities and enhanced performance achieved under these greener and milder reaction conditions cannot be replicated using traditional Schlenk-type synthetic techniques with dry *VOCs*. This unique outcome and acceleration of *s*-block metal-mediated reactions in strongly associated reaction media (such as water and *DESs*) appear to be driven by a combination of factors. These encompass (*i*) intricate dynamic equilibria and changes in the aggregation states of organometallic species, (*ii*) the presence of robust H-bonded networks, (*iii*) the formation of highly efficient “*ate*” complexes exhibiting enhanced nucleophilicity, and (*iv*) catalytic effects occurring at the interface between organic compounds and the aforementioned green solvents (water and *DESs*) [21,22,23,24,25,26,27]. In this line, a recent computational investigation carried out by Koszinowski and Rahrt, aimed at addressing the experimental observation of how organozinc reagents survive instantaneous protonolysis in protic media, also disclosed the significance of achieving favorable kinetics despite less favorable thermodynamics [104].

For sure, the structure and dynamical properties of water and *DESs* are more complex than expected, and reactions of polar organometallic compounds of *s*-block elements across the “oil–water/*DES*” interface may represent another example of *compartmentalization* in organic synthesis [105]. Indeed, the use of *I*nterphase-*R*ich *A*queous *S*ystems (*IRAS*) are known to promote and guide organic reactions [106]. A deeper understanding of phenomena taking place at such an interface, when polar organometallics meet water or *DESs*, will enable the full exploitation of this chemistry for novel and exciting transformations, working under conditions that no one would have ever envisioned adopting before!

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
