# Peer review of "Recent Advancements in the Utilization of s-Block Organometallic Reagents in Organic Synthesis with Sustainable Solvents"

_molecules, 2024, doi:10.3390/molecules29071422_

Round 1

Reviewer 1 Report

Comments and Suggestions for Authors

This is a short review on  the utilization of highly polar s-block organometallic reagents  in organic synthesis under  air/moisture, at room temperature,  using  sustainable solvents as reaction media. The review is very good  and can be  published in the journal. It would be very useful for the readers  and can improve the review if  the authors can comment on the  the following:

1) Why CPME or 2-MeTHF  were used as more suitable solvent.

2) Authors should write  RNa  after RLi and then RMgX .

3) The authors can  also include  a brief description of DESs solvents.

4) In the 2.2 heading, yields of the reaction should be presented in the paragraph.

5) The reactions shown in Scheme 2 , should mention the yields of the reaction.

Overall the review is publishable.

Author Response

We are delighted that it has been well received by the two reviewers and we are very grateful for their useful comments. Following their recommendations, the manuscript has now been revised appropriately (changes highlighted in yellow), addressing all their comments in full as follows:

■ Reviewer 1

This is a short review on the utilization of highly polar s-block organometallic reagents in organic synthesis under air/moisture, at room temperature, using sustainable solvents as reaction media. The review is very good and can be published in the journal. It would be very useful for the readers and can improve the review if the authors can comment on the following:

Answer: We are grateful to this referee for his/her positive comments, and for this summary of our work.

1) Why CPME or 2-MeTHF were used as more suitable solvent

Answer: As suggested, we have included a new paragraph in the introduction of the manuscript to clarify this point:

- Introduction; page 2; lines 74-83: In this sense, 2-MeTHF is considered a biorenewable biomass-derived solvent as it can be produced from furfural without the need for petrochemical protocols [35]. Moreover, the immiscibility between water and 2-MeTHF permits its straightforward employment in liquid–liquid extraction (avoiding the use of toxic VOC solvents for the purification/isolation of the desired final organic products), thus opening the door to its use as an alternative solvent in organometallic-mediated organic transformations [36]. On the other hand, CPME is also considered a promising eco-friendly solvent due to its intrinsic valuable properties such as low peroxide formation rate, stability under basic and acidic conditions, and relatively high boiling point [37].

2) Authors should write RNa after RLi and then RMgX

Answer: We totally agree with the comments of this reviewer. As suggested, we have written RNa after RLi in the main text of the manuscript:

- Abstract, page 1, lines 14-15: … (specifically, RLi, RNa and RMgX, compounds)

- Keywords, page 1: Keywords: Organolithium; Organosodium; Organomagnesium

- Introduction; page 1, 1st paragraph: …organolithium (RLi) [1-5], organosodium (RNa) [6-9] and organomagnesium (RMgX or R2Mg) [10-12] compounds

- Section 6, page 14, 1st paragraph: … The findings presented in this review challenge the conventional notion that organolithium (RLi), organosodium (RNa) or Grignard (RMgX)…

- Conclusions, page 17, 1st line: Polar organometallic reagents, belonging to the s-block elements (RLi, RNa or RMgX)

- References, page 19: we have changed the numbering of the reviews/chapters related with RNa and RMgX chemistry:

3) The authors can also include a brief description of DESs solvents.

Answer: As suggested, we have included a new paragraph in the introduction of the manuscript to clarify this point:

- page 2, lines 62-66: i) in the presence of air; ii) at room temperature; and iii) in sustainable, protic, non-toxic, and non-dried solvents, such as water or Deep Eutectic Solvents (DESs) which are eutectic mixtures of Lewis or Brønsted acids and bases strongly associated with each other, exhibiting a significant depression of freezing points far below that of ideal mixtures (each component has a higher melting point than the mixture) [28].

4) In the 2.2 heading, yields of the reaction should be presented in the paragraph.

Answer: We totally agree with the comments of this reviewer. As suggested, we have included the reaction yields:

- page 5; line 196: achieving yield values up to 98%.

5) The reactions shown in Scheme 2, should mention the yields of the reaction.

Answer: Again, we totally agree with the comments of this reviewer. As suggested, we have introduced the yields of the reaction in Scheme 2:

Overall the review is publishable.

Answer: Again, we thank warmly this referee for his/her positive comments.

■ Reviewer 2

The paper is interesting and well written. It shows an unusual behaviour of s-block elements in the presence of DES air and water.

Answer: We thank warmly this referee for his/her positive comments.

It would be desirable to discuss more about the reason of such a behaviour. A short statement at the end of Conclusions seems a bit unsatisfactory.

Answer: We agree with the comment of this reviewer. As suggested, we have included a new paragraph in the conclusion of the manuscript to clarify this point:

- Conclusions, page 18, lines 654-668: The examples reviewed not only underscore the sustainability of the transformations, but also reveal that the exceptional chemoselectivities and enhanced performance achieved under these greener and milder reaction conditions cannot be replicated using traditional Schlenk-type synthetic techniques with dry VOCs. This unique outcome and acceleration of s-block metal-mediated reactions in strongly associated reaction media (such as water and DESs) appear to be driven by a combination of factors. These encompass i) intricate dynamic equilibria and changes in the aggregation states of organometallic species; ii) the presence of robust H-bonded networks; iii) the formation of highly efficient "ate" complexes exhibited enhanced nucleophilicity; and iv) catalytic effects occuring at the interface between organic compounds and the aforementioned green solvents (water and DESs) [21-27]. In this line, a recent computational investigation carried out by Koszinowski and Rahrt, aimed at addressing the experimental observation of how organozinc reagents survive instantaneous protonolysis in protic media, also disclosed the significance of achieving favorable kinetics despite to a less favorable thermodynamics [105].

  1. It would be advisable to enlist the abbreviations used. Although they are explained in the text, it is necessary to return to find them.

Answer: We thank the reviewer for this suggestion. Following his/her advice, we have introduced an abbreviation list at the final part of the main text of the manuscript (pages 18-19):

  1. Line 183: imines are not synthesized here but used as reagents. 3. Line 545: the correct name of the compound should read: methyl p-formylbenzoate

Answer: We thank the reviewer for spotting these two mistakes. Both have been corrected.

To sum up, the paper deserves publication after minor revision.

Answer: Again, we thank warmly this referee for his/her positive comments.

Finally, we have introduced in our review a very recent paper (published in March 2024) related with the use of continuous on-water reaction conditions in the addition of RLi reagents into imines. A new paragraph and a new reference has been introducing in the main text of the manuscript:

- Page 5, 1st paragraph: Recently (in 2024), Benaglia and co-workers have advanced this field by developing a highly efficient and rapid (10–20 s), continuous on-water organolithium addition to imines. This innovative approach allows for the synthesis of functionalized secondary amines in high yields (up to 97%), effectively addressing safety concerns associated with the process. To achieve these impressive results, the authors employed continuous stirred tank reactors (CSTRs) ensuring a rapid stirring rate. Notably, within a single 2.5 mL CSTR, 5 g of amine can be synthesized in just 3 min. Furthermore, the methodology has been successfully utilized for the synthesis of an enantiomerically enriched chiral amine (98% enantiomeric excess) [81].

- page 21, reference [81]: Brucoli, J.; Puglisi, A.; Rossi, S.; Gariboldi, D.; Brenna, D.; Maule, I.; Benaglia, M. A three-minute gram-scale synthesis of aminesvia ultrafast ‘‘on-water’’ in continuo organolithium addition to imines. Cell. Rep. Phys. Sci. 2024, 5, 101838.

We hope that you find the revised manuscript to now be suitable for publication in Molecules.

Looking forward to hearing from you in due course.

Reviewer 2 Report

Comments and Suggestions for Authors

The paper is interesting and well written. It shows an unusual behavior of
s-block elements in the presence of DES air and water. It would be desirable to discuss more about the reason of such a behvaviour. A short statement at the end of Conclusions seems a bit usatisfactory.

Some minor reservations:

1. It would be advisable to enlist the abbreviations used . Although they are explained in the text, it is necessary to return to find them.

2. Line 183: imines are not synthesized here but used as reagents

3. Line 545: the correct name of the compound should read:
methyl p-formylbenzoate

To sum up, the paper deserves publication after minor revision.

Author Response

We are delighted that it has been well received by the two reviewers and we are very grateful for their useful comments. Following their recommendations, the manuscript has now been revised appropriately (changes highlighted in yellow), addressing all their comments in full as follows:

■ Reviewer 1

This is a short review on the utilization of highly polar s-block organometallic reagents in organic synthesis under air/moisture, at room temperature, using sustainable solvents as reaction media. The review is very good and can be published in the journal. It would be very useful for the readers and can improve the review if the authors can comment on the following:

Answer: We are grateful to this referee for his/her positive comments, and for this summary of our work.

1) Why CPME or 2-MeTHF were used as more suitable solvent

Answer: As suggested, we have included a new paragraph in the introduction of the manuscript to clarify this point:

- Introduction; page 2; lines 74-83: In this sense, 2-MeTHF is considered a biorenewable biomass-derived solvent as it can be produced from furfural without the need for petrochemical protocols [35]. Moreover, the immiscibility between water and 2-MeTHF permits its straightforward employment in liquid–liquid extraction (avoiding the use of toxic VOC solvents for the purification/isolation of the desired final organic products), thus opening the door to its use as an alternative solvent in organometallic-mediated organic transformations [36]. On the other hand, CPME is also considered a promising eco-friendly solvent due to its intrinsic valuable properties such as low peroxide formation rate, stability under basic and acidic conditions, and relatively high boiling point [37].

2) Authors should write RNa after RLi and then RMgX

Answer: We totally agree with the comments of this reviewer. As suggested, we have written RNa after RLi in the main text of the manuscript:

- Abstract, page 1, lines 14-15: … (specifically, RLi, RNa and RMgX, compounds)

- Keywords, page 1: Keywords: Organolithium; Organosodium; Organomagnesium

- Introduction; page 1, 1st paragraph: …organolithium (RLi) [1-5], organosodium (RNa) [6-9] and organomagnesium (RMgX or R2Mg) [10-12] compounds

- Section 6, page 14, 1st paragraph: … The findings presented in this review challenge the conventional notion that organolithium (RLi), organosodium (RNa) or Grignard (RMgX)…

- Conclusions, page 17, 1st line: Polar organometallic reagents, belonging to the s-block elements (RLi, RNa or RMgX)

- References, page 19: we have changed the numbering of the reviews/chapters related with RNa and RMgX chemistry:

3) The authors can also include a brief description of DESs solvents.

Answer: As suggested, we have included a new paragraph in the introduction of the manuscript to clarify this point:

- page 2, lines 62-66: i) in the presence of air; ii) at room temperature; and iii) in sustainable, protic, non-toxic, and non-dried solvents, such as water or Deep Eutectic Solvents (DESs) which are eutectic mixtures of Lewis or Brønsted acids and bases strongly associated with each other, exhibiting a significant depression of freezing points far below that of ideal mixtures (each component has a higher melting point than the mixture) [28].

4) In the 2.2 heading, yields of the reaction should be presented in the paragraph.

Answer: We totally agree with the comments of this reviewer. As suggested, we have included the reaction yields:

- page 5; line 196: achieving yield values up to 98%.

5) The reactions shown in Scheme 2, should mention the yields of the reaction.

Answer: Again, we totally agree with the comments of this reviewer. As suggested, we have introduced the yields of the reaction in Scheme 2:

Overall the review is publishable.

Answer: Again, we thank warmly this referee for his/her positive comments.

■ Reviewer 2

The paper is interesting and well written. It shows an unusual behaviour of s-block elements in the presence of DES air and water.

Answer: We thank warmly this referee for his/her positive comments.

It would be desirable to discuss more about the reason of such a behaviour. A short statement at the end of Conclusions seems a bit unsatisfactory.

Answer: We agree with the comment of this reviewer. As suggested, we have included a new paragraph in the conclusion of the manuscript to clarify this point:

- Conclusions, page 18, lines 654-668: The examples reviewed not only underscore the sustainability of the transformations, but also reveal that the exceptional chemoselectivities and enhanced performance achieved under these greener and milder reaction conditions cannot be replicated using traditional Schlenk-type synthetic techniques with dry VOCs. This unique outcome and acceleration of s-block metal-mediated reactions in strongly associated reaction media (such as water and DESs) appear to be driven by a combination of factors. These encompass i) intricate dynamic equilibria and changes in the aggregation states of organometallic species; ii) the presence of robust H-bonded networks; iii) the formation of highly efficient "ate" complexes exhibited enhanced nucleophilicity; and iv) catalytic effects occuring at the interface between organic compounds and the aforementioned green solvents (water and DESs) [21-27]. In this line, a recent computational investigation carried out by Koszinowski and Rahrt, aimed at addressing the experimental observation of how organozinc reagents survive instantaneous protonolysis in protic media, also disclosed the significance of achieving favorable kinetics despite to a less favorable thermodynamics [105].

  1. It would be advisable to enlist the abbreviations used. Although they are explained in the text, it is necessary to return to find them.

Answer: We thank the reviewer for this suggestion. Following his/her advice, we have introduced an abbreviation list at the final part of the main text of the manuscript (pages 18-19):

  1. Line 183: imines are not synthesized here but used as reagents. 3. Line 545: the correct name of the compound should read: methyl p-formylbenzoate

Answer: We thank the reviewer for spotting these two mistakes. Both have been corrected.

To sum up, the paper deserves publication after minor revision.

Answer: Again, we thank warmly this referee for his/her positive comments.

Finally, we have introduced in our review a very recent paper (published in March 2024) related with the use of continuous on-water reaction conditions in the addition of RLi reagents into imines. A new paragraph and a new reference has been introducing in the main text of the manuscript:

- Page 5, 1st paragraph: Recently (in 2024), Benaglia and co-workers have advanced this field by developing a highly efficient and rapid (10–20 s), continuous on-water organolithium addition to imines. This innovative approach allows for the synthesis of functionalized secondary amines in high yields (up to 97%), effectively addressing safety concerns associated with the process. To achieve these impressive results, the authors employed continuous stirred tank reactors (CSTRs) ensuring a rapid stirring rate. Notably, within a single 2.5 mL CSTR, 5 g of amine can be synthesized in just 3 min. Furthermore, the methodology has been successfully utilized for the synthesis of an enantiomerically enriched chiral amine (98% enantiomeric excess) [81].

- page 21, reference [81]: Brucoli, J.; Puglisi, A.; Rossi, S.; Gariboldi, D.; Brenna, D.; Maule, I.; Benaglia, M. A three-minute gram-scale synthesis of aminesvia ultrafast ‘‘on-water’’ in continuo organolithium addition to imines. Cell. Rep. Phys. Sci. 2024, 5, 101838.

We hope that you find the revised manuscript to now be suitable for publication in Molecules.

Looking forward to hearing from you in due course.

Best wishes

Prof. Vito Capriati and Dr. Joaquín García-Álvarez
